# Comprehensive Comparison between Vision Transformers and Convolutional Neural Networks for Face Recognition Tasks

## Abstract

This paper presents a comprehensive comparison between Vision Transformers and Convolutional Neural Networks for face recognition related tasks, including extensive experiments on the tasks of face identification and verification. Our study focuses on six state-of-the-art models: EfficientNet, Inception, MobileNet, ResNet, VGG, and Vision Transformers. Our evaluation of these models is based on five diverse datasets: Labeled Faces in the Wild, Real World Occluded Faces, Surveillance Cameras Face, UPM-GTI-Face, and VGG Face 2. These datasets present unique challenges regarding people diversity, distance from the camera, and face occlusions such as those produced by masks and glasses. Our contribution to the field includes a deep analysis of the experimental results, including a thorough examination of the training and evaluation process, as well as the software and hardware configurations used. Our results show that Vision Transformers outperform Convolutional Neural Networks in terms of accuracy and robustness against distance and occlusions for face recognition related tasks, while also presenting a smaller memory footprint and an impressive inference speed, rivaling even the fastest Convolutional Neural Networks. In conclusion, our study provides valuable insights into the performance of Vision Transformers for face recognition related tasks and highlights the potential of these models as a more efficient solution than Convolutional Neural Networks.

## 1 Introduction

The field of computer vision has seen a remarkable evolution in recent years, with deep learning techniques playing a crucial role in its advancement (Chai et al. (2021)). One of the most recent and promising developments in this field is the introduction of the Vision Transformers (ViTs) in 2020 (Dosovitskiy et al. (2020)). ViTs are a type of artificial neural network that have shown remarkable results in generic image classification tasks, surpassing previous state-of-the-art results on many benchmark datasets (Khan et al. (2022); Han et al. (2022)). Since the appearance of the ViTs, several studies have been published comparing them with the most outstanding Convolutional Neural Networks (CNNs) in generic image classification tasks (Raghu et al. (2021); Guo et al. (2022); Benz et al. (2021)). These studies have concluded that ViTs not only outperform CNNs on accuracy, but also have higher shape bias, and are largely more consistent with human errors (Tuli et al. (2021)).

In computer vision, face recognition is a field of great interest, with a wide range of applications, including security, biometrics, and human-computer interaction (Du et al. (2022); Wang & Deng (2021)). The motivation behind our investigation into facial recognition as the primary task stems from the evolving landscape of computer vision and the ongoing paradigm shift between CNNs and ViTs (Maurício et al. (2023)). While previous studies have presented very specific face recognition solutions using CNNs (Taigman et al. (2014); Sun et al. (2014); Chen et al. (2018); Martindez-Diaz et al. (2019); Yan et al. (2019)), ViTs (Zhong & Deng (2021); Sun & Tzimiropoulos (2022)), and hybrids (George et al. (2023); Li et al. (2023)), our focus lies in presenting a comparative analysis between ViTs and CNNs within this domain. CNNs have historically been the cornerstone of face recognition models; however, with the emergence of ViTs and their potential advantages, a comprehensive evaluation becomes imperative. Face recognition presents very specific features and challenges, notably related to the low inter-class variance and the high intra-class variance observed in most face image datasets (Cao et al. (2018); Huang et al. (2008)). This complexity makes face recognition a more demanding task than generic image classification and, therefore, an ideal test bed for evaluating the performance of ViTs.

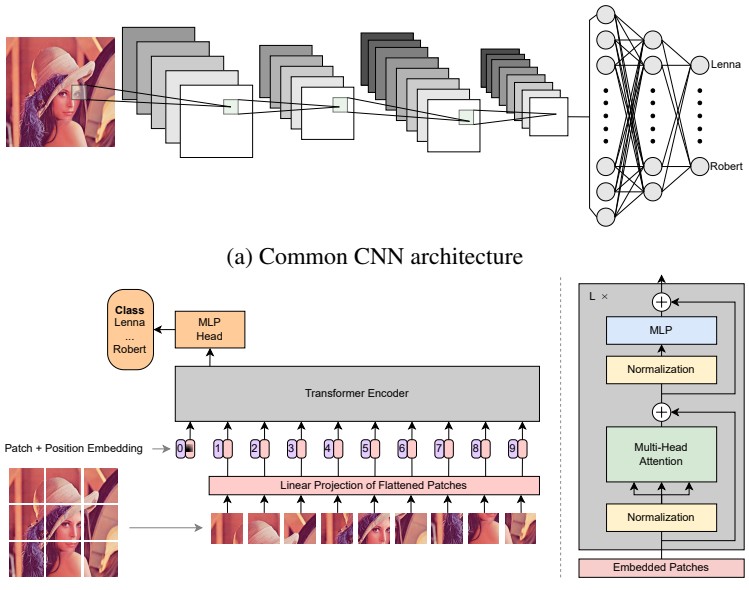

Figure 1: Visual depiction of the fundamental differences between Vision Transformers (a) and Convolutional Neural Networks (b) architectures and operational mechanisms.

In this paper, we analyze the performance of ViTs in face recognition tasks (identification and verification), using five different datasets that allow assessing several types of challenges. The results of these analysis provide valuable insights into the performance of ViTs in this specific domain and contribute to a better understanding of the suitability of ViTs for facial recognition tasks. In addition, to determine to what extent the conclusions from prior studies on generic image classification tasks are extensible to face recognition tasks, we compare ViTs with some of the most outstanding CNNs in the field of face recognition.

## 2   ViT FOR FACE RECOGNITION

The field of face recognition can be broadly classified into two sub-tasks: face identification and face verification (Du et al. (2022)). Face identification involves the determination of whether a particular face image corresponds to a person in a given dataset and can be treated as a one-to-many problem. In contrast, face verification involves determining the similarity between two face images, and is a one-to-one problem. Face identification differs from face verification in that the former can be formulated as a classification problem and so, has prior knowledge of the number of classes, whereas the latter does not. This influences the selection of appropriate loss functions for each task.

For both tasks, face embeddings, also known as face descriptors, are high-level representations of a face image obtained in the final layers of a Deep Neural Network (DNN). Ideally, these embeddings should have low intra-class variance and high inter-class variance, meaning that they are similar for images of the same subject and distinct for images of different subjects. In the case of face identification, this is typically achieved implicitly through a softmax loss function, while in face verification, it is usually achieved explicitly through the use of a different loss function (e.g., Triplet-loss (Schroff et al. (2015)), ArcFace (Deng et al. (2019)), CosFace (Wang et al. (2018)), or SphereFace (Liu et al. (2017))).

### 2.1   DIFFERENCES BETWEEN ViTs AND CNNs

The operational mechanisms of ViTs diverges significantly from that of CNNs, as illustrated in Fig. 1. CNNs operate through a series of layers, each of which detects different features in an image by applying various filters, and sends them to the subsequent layer. These filters can initially begin detecting very simple features, and increase in complexity to detect features that uniquely identify a specific object or category.

In contrast, ViTs break an image into patches, which are flattened and turned into a series of tokens through a linear projection to enable them to be fed into the network (originally developed for Natural Language Processing applications, and thus designed to accept word-like inputs). These tokens are passed through a series of transformer encoder layers, which exhibit a remarkable capacity for understanding how different parts of the picture relate to each other. This is achieved by means of a multi-head self-attention mechanism. Each head learns diverse dependencies by generating three representations (query, key, and value) for each input token, which are then processed to obtain an output representation. These output representations are subsequently transmitted to the next transformer encoder layer. These representations are similar to the activation maps outputted by the filters of a CNN in that they serve to detect features that uniquely define a specific object or category, but there are some significant differences.

One of the major differences between ViTs and CNNs lies in the large field of view of the initial ViTs' layers. CNNs employ a fixed-size kernel field of view, gradually extending it by repeatedly convolving the information around the kernel layer by layer. In contrast, ViTs utilize a self-attention mechanism that enables the model to have a whole field of view, even at the lowest layer. Hence, ViTs obtain global representations from the beginning, while CNNs need to propagate layers to obtain global representations. This can also be a disadvantage for ViTs, which require large amounts of data to obtain local representations in the lowest layers. However, this can be addressed through a well-designed pre-training strategy.

ViTs and CNNs also differ significantly in terms of their memory footprint (Mangalam et al. (2022)). During training, CNNs must save all intermediate activation maps resulting from the performed convolutions. These activation maps have a significant impact on the memory footprint during training, and can quickly restrict the maximum number of samples in a batch, which in turn can affect the ability of the model to converge. ViTs, on the other hand, are much less affected by this issue, as the initial step divides the image into tokens, which have a much smaller memory footprint than activation maps during training.

## 3 EXPERIMENTS

We conduct a comparative analysis of the performance of ViTs with some of the most widely used CNNs, namely: ResNet (He et al. (2016)), VGG (Simonyan & Zisserman (2014)), Inception (Szegedy et al. (2015)), MobileNet (Howard et al. (2017)), and EfficientNet (Tan & Le (2019)). More specifically, we compare ViT_B32 (ViT base model with a patch size of 32 pixels) with ResNet_50, VGG_16, Inception_V3, MobileNet_V2, and EfficientNet_B0, although for simplicity we will refer to them as ViT, ResNet, VGG, Inception, MobileNet, and EfficientNet. To gain comprehensive insights into their respective strengths and limitations, the six networks have been tested both during training and evaluation stages. Additionally, to ensure a rigorous and objective comparison, they have been trained under standard hyperparameter settings commonly used in literature: an image size of 224, batch size of 256, 25 epochs, Adam optimizer, and a learning rate of 0.0001. These hyperparameters were selected to maintain consistency and neutrality across the models, ensuring a fair evaluation and avoiding any bias towards optimizing a specific architecture. This uniformity in training conditions simplifies the subsequent comparative analysis, allowing us to make more meaningful and objective comparisons among ViTs and CNNs. It is important to emphasize that while this approach streamlines the comparison process, we remain aware that, in practice, networks might indeed perform optimally with distinct hyperparameter settings tailored to their unique requirements. Furthermore, we included fixed seed settings, particularly across diverse datasets, to provide a more robust evaluation framework, ensuring the reliability and stability of our comparative findings.

The hardware used consisted of a workstation powered by an Intel i9 13900K CPU, two NVIDIA RTX 4090 GPU, and 128 GB of RAM running on a Ubuntu 22.04 operating system. The six networks were implemented in Python programming language, using the TensorFlow framework and a data parallelization strategy to split batches across both GPUs. We have made the implementation of the experiments publicly available[1].

---

[1]Link omitted for the double-blind review process

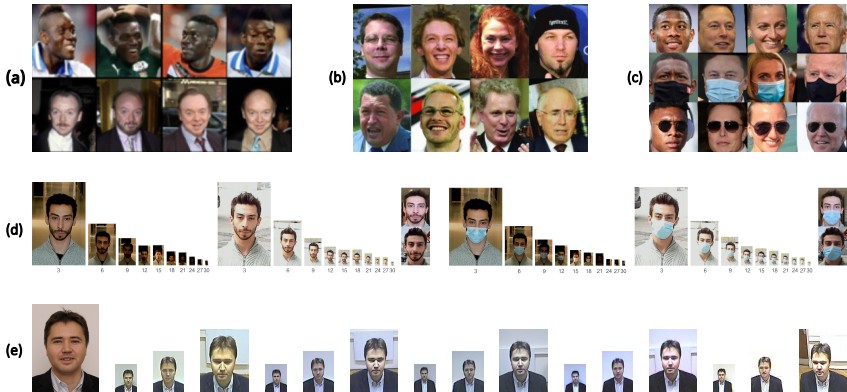

Figure 2: Sample images from the VGG Face 2 (a), LFW (b), ROF (c), UPM-GTI-Face (d), and SCface (e) datasets used in our experiments.

## 3.1 DATASETS

In this study, we have utilized five distinct facial image datasets, each possessing unique characteristics as described in the subsequent paragraphs.

We have used VGG Face 2 (Cao et al. (2018)) dataset mainly for training purposes. This dataset consists of 3.31 million images of $9,131$ subjects, with an average of 363 images for each subject (see Fig. 2a). The large number of images and subjects allowed us to train the six different networks covering a large range of pose, age, illumination, and ethnicity. We have used $90\%$ of the images for training (2.83 million) and $5\%$ ($157,000$) for validation during training. The remaining $5\%$ has been used for evaluation purposes, setting aside $157,000$ random images to measure the face identification performance of the networks.

We have also used the popular Labeled Faces in the Wild (LFW) (Huang et al. (2008)) dataset for the evaluation of face verification performance of the six networks. This dataset is a public benchmark for face verification in unconstrained conditions, exhibiting a large variation in pose, illumination, facial expression, age, gender, and race, as illustrated in Fig. 2b. It contains more than $13,000$ face images of more than $5,700$ subjects, from which $1,680$ subjects have two or more distinct photos. For our evaluation, we followed the guidelines provided by the dataset's authors, resulting in the formation of 6,000 pairs of faces. Among these pairs, $3,000$ correspond to correct pairings, while the remaining $3,000$ correspond to incorrect pairings.

The Real-World Occluded Faces (ROF) (Erakın et al. (2021)) dataset has been used to assess the face verification performance of the networks. This dataset comprises facial images featuring genuine upper-face and lower-face occlusions caused by sunglasses and face masks, respectively (refer to Fig. 2c). The dataset encompasses a total of $6,421$ neutral face images, $4,627$ face images with sunglasses, and 678 face images with masks. There are 47 subjects with neutral, masked, and sunglasses images, 114 subjects with neutral and sunglasses images, while 20 subjects have only neutral and masked images. The identities within the dataset are drawn from a pool of celebrities and politicians, and all the images are sourced from real-life situations, exhibiting substantial variations in pose and illumination. Pairs within this dataset were formed by matching each neutral image with all the images of subjects wearing either masks or sunglasses.

The UPM-GTI-Face (Rodrigo et al. (2022)) dataset has been utilized for the evaluation of face verification performance of the networks. This dataset serves as a publicly available benchmark for evaluating the robustness of face recognition networks under challenging surveillance scenarios, notably for assessing the join impact of distance and face masks. The dataset encompasses a total of 484 images of 11 subjects, acquired at 10 annotated distances ranging from 3 to 30 meters, spanning both indoor and outdoor environments, and for 2 mask conditions (with and without). For every combination of subject, environment, and mask condition, the dataset provides a mugshot gallery image at a standard distance of 1 meter, accompanied by 10 probe images, as shown in Fig. 2d. To ensure comprehensive coverage of the face verification task, pairs were meticulously formed by mixing indoor and outdoor environments, while keeping the masked and unmasked experiments

separate. Consequently, for each of the 10 intermediate distances (involving 2 environments and 11 subjects at each intermediate distance), the dataset contains 22 mugshot gallery images and 22 probe images. This arrangement results in the creation of 484 image pairs for every distance, with only 44 of these pairs representing correct matches for each mask condition (4 correct pairs per subject).

SCface (Grgic et al. (2011)) database has served to evaluate the performance of face verification. This dataset comprises static images of human faces captured in an uncontrolled indoor environment using five video surveillance cameras of varying qualities, as depicted in Fig. 2e. In total, the database encompasses facial images of 130 different subjects. Each subject within the database is represented by a mugshot gallery image and 15 probe images, which were captured using 5 distinct cameras and at 3 different distances: close, medium, and long range. The utilization of cameras with varying image qualities replicates real-world conditions and facilitates the evaluation of robust face recognition algorithms, particularly in scenarios related to law enforcement and surveillance. Pairs for evaluation were established by comparing each mugshot gallery image with all the available probe images within the dataset.

## 3.2 Evaluation Metrics

Different evaluation metrics have been used during the experimental evaluation of the networks. For the task of face identification, we have used accuracy (to measure how often predictions match their targets) and top-5 accuracy (to measure how often targets are within the top 5 predictions).

For the task of face verification, we have used receiver operating characteristics (ROC) curves, as well as the area under their curves (AUC) and the equal error rate (EER). ROC curves illustrate the ability of a binary classifier (i.e., whether there is a match or not) as its discrimination threshold is varied, resulting in false positive rate (FPR) and true positive rate (TPR) metrics for every threshold. The AUC of each ROC serves as a performance measurement that represents the degree of separability between classes and is bounded between 0 (the worst measure of performance) and 1 (a perfect measure of performance), with 0.5 indicating that a network has no class separation capability whatsoever. The EER represents the point in a ROC curve where the FPR matches the false negative rate (FNR), that is, the point at which the diagonal line going from the top-left to the bottom-right crosses the ROC curve.

## 3.3 Training

To speed up convergence during training, networks were pre-trained using the Imagenet (Deng et al. (2009)) large-scale dataset. Subsequently, we conducted training on the VGG Face 2 dataset, utilizing both its training and validation subsets. The outcomes of this training endeavor are summarized in Table 1, which provides the highest accuracy achieved on both the training and validation sets. Remarkably, all six networks demonstrated exceptional performance, with accuracy levels approaching 100% on both the training and validation sets by the conclusion of the training process. Notably, ViT stood out not only by achieving the highest validation results but also by accomplishing this feat in fewer epochs than its CNN counterparts (for further details, please refer to the additional material in our publicly available implementation, accessible through the provided link). Furthermore, upon completing training, ViT results on the validation set were still superior to those of the training set by a large margin. Specifically, ViT's accuracy rose from 98.86% to 99.81%, reflecting a remarkable 83.33% loss reduction. This indicates that overfitting (Webster et al. (2019)) has not yet occurred and as such, it is possible that the results could be further improved if the training continued for a few more epochs. On the other hand, CNNs started to exhibit some overfitting signs as the training concluded, raising concerns about their ability to perform as good on datasets that differ from the one they were trained on.

## 3.4 Evaluation

The 157, 000 images from the VGG Face 2 dataset that were excluded from the training process served as a valuable resource for evaluating the face identification capabilities of the six networks under consideration. In Table 2, we present the accuracy results for each network when classifying these images as belonging to one of the subjects within the dataset, alongside the corresponding top-5 accuracy scores. Additionally, we provide information regarding the number of parameters

Table 1: Training summary on the VGG Face 2 dataset. The accuracy corresponds to the highest values obtained on the training and validation sets during training for the face identification task.

| Network | Training Accuracy | Validation Accuracy |
|---|---|---|
| ViT_B32 | 98.86% | **99.81%** |
| ResNet_50 | **99.66%** | 99.55% |
| VGG_16 | 97.59% | 98.21% |
| Inception_V3 | 99.46% | 99.54% |
| MobileNet_V2 | 99.18% | 98.90% |
| EfficientNet_B0 | 99.27% | 95.90% |

Table 2: Face identification results obtained on the evaluation set of VGG Face 2 dataset. Here we report the test accuracy and top-5 accuracy, as well as the number of parameters of each model, accompanied by the inference time per batch of 256 images.

| Network | Test Accuracy | Top-5 Test Accuracy | # Parameters | Inference Time |
|---|---|---|---|---|
| ViT_B32 | **99.80%** | **100%** | 94 M | 78 ms/batch |
| ResNet_50 | 99.57% | 99.99% | 41 M | 105 ms/batch |
| VGG_16 | 98.17% | 99.91% | 170 M | 141 ms/batch |
| Inception_V3 | 99.58% | 99.99% | 39 M | 64 ms/batch |
| MobileNet_V2 | 98.92% | 99.96% | 13 M | **63** ms/batch |
| EfficientNet_B0 | 95.90% | 99.42% | 15 M | 91 ms/batch |

and the inference time per batch. The latter denotes the time required for processing a batch of 256 images when the networks are not in training mode. The findings from this evaluation clearly underscore the superiority of ViTs in terms of accuracy. ViT not only scores the highest accuracy but also distinguishes itself as the only network to attain a flawless 100% top-5 accuracy. Furthermore, ViT's inference speed is notably competitive, aligning with some of the fastest CNNs, with MobileNet being the fastest one. Remarkably, ViT inference speed is only 23.81% slower than that of MobileNet, despite having more than 7 times the number of parameters.

The ROC curves presented in Figure 3 offer a summary of the outcomes derived from the face verification task conducted on the LFW dataset by the six networks under scrutiny. While all six networks exhibit commendable performance, ViT stands out with a slightly superior performance, positioning closer to the top-left corner of the graph. Notably, ViT achieves the highest AUC value and the lowest EER. It is worth noting that the LFW dataset, being an older dataset, does not pose a substantial challenge, and all six networks demonstrate exceptional performance on it. Nevertheless, this dataset serves a crucial purpose by showcasing that all six networks are capable of extracting high-quality facial embeddings suitable for face verification tasks. The forthcoming datasets, however, present more formidable challenges.

The outcomes of the face verification task conducted on the SCface dataset are visually depicted in Fig. 4. Within this dataset, each subject is represented by a high-quality mugshot gallery image, alongside several images captured by five different cameras at three distinct distances. The ROC curves showcased in Figure 4 result from comprehensive comparisons, encompassing all gallery images against all probe images. Specifically, Figure 4a presents the outcomes of comparing all mugshot gallery images against the probe images captured at the long distance by every camera. Likewise, Fig. 4b and Fig. 4c offer parallel insights for the medium and close distances, respectively. To consolidate these findings, Figure 4d synthesizes all the results into a single representation. From these results, it becomes evident that while all six networks perform reasonably well at close distance ViT significantly outperforms its CNN counterparts at medium and long distances. This observation suggests that ViT's face embeddings demonstrate greater robustness to variations in distance in the context of face recognition tasks.

The results obtained for the face verification task in the ROF dataset are presented in Fig. 5. In Fig. 5a, we present a histogram illustrating the dataset's distribution, revealing the number of images each subject in the dataset has for each category, namely: neutral, masked, and sunglasses. Figures 5b, 5c, and 5d delve into the outcomes obtained when comparing neutral images of every subject against images with masks, sunglasses, and both categories combined, respectively. These

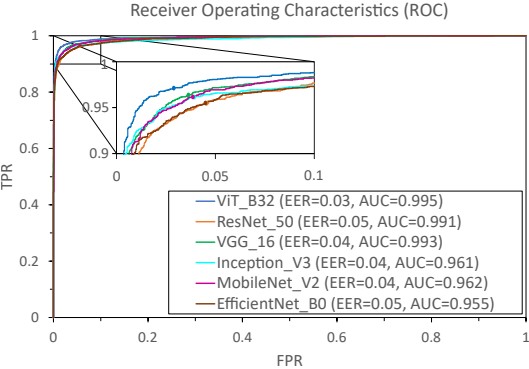

Figure 3: ROC curves obtained for the pairs of images proposed in the LFW dataset. AUC scores as well as EERs are displayed in the legend for each network.

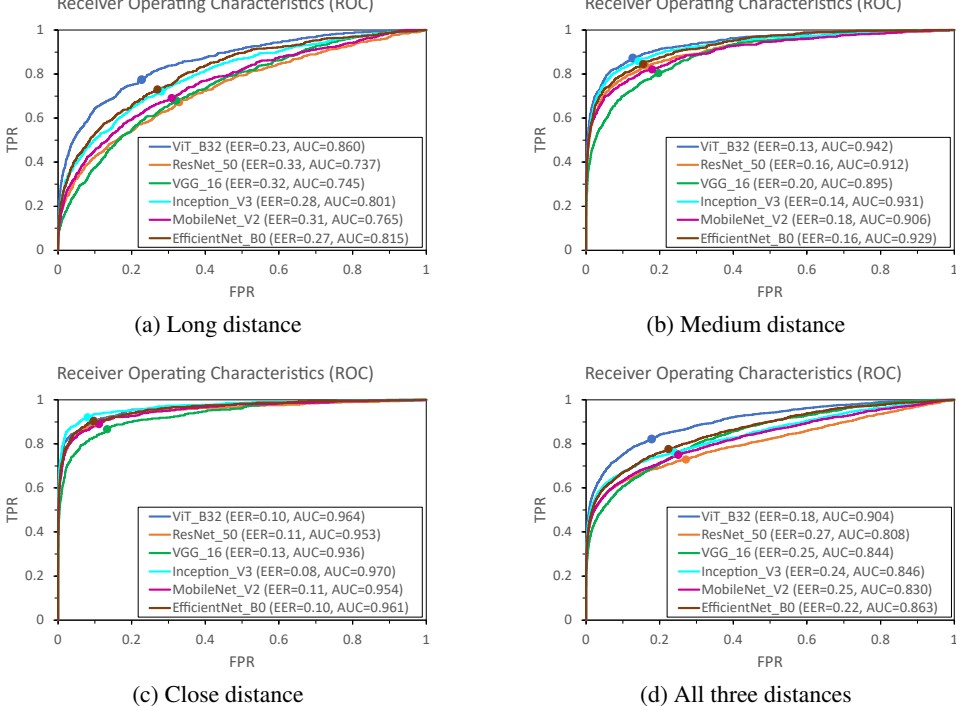

Figure 4: ROC curves obtained for SCface dataset. This figure encompasses the results obtained for long (a), medium (b), and close (c) distances, as well as the results obtained when synthesizing all results into a single representation.

visualizations distinctly showcase ViT's superior performance in handling occlusions compared to CNNs. ViT excels in deriving global face embeddings, whereas CNNs demonstrate proficiency in extracting local representations. This is likely the reason why ViT presents itself as a more robust approach against occlusion in face recognition tasks, as it is less affected by local occlusions, whether they occur in the lower or upper facial regions.

The results obtained for the face verification task in the UPM-GTI-Face dataset are presented in Fig. 6. This figure provides a summary of the information contained in the ROC curves computed for every intermediate distance, using the AUC values. In the absence of masks, as depicted in Figure 6a, the performance of the six networks is similar for distances ranging from 3 to 12 meters, while for distances greater than 15 meters, the performance of all networks experiences a significant decline. Notably, for distances from 6 meters onwards, ViT consistently outperforms CNNs, as evidenced by its superior AUC scores. This suggests that ViT's face embeddings exhibit greater

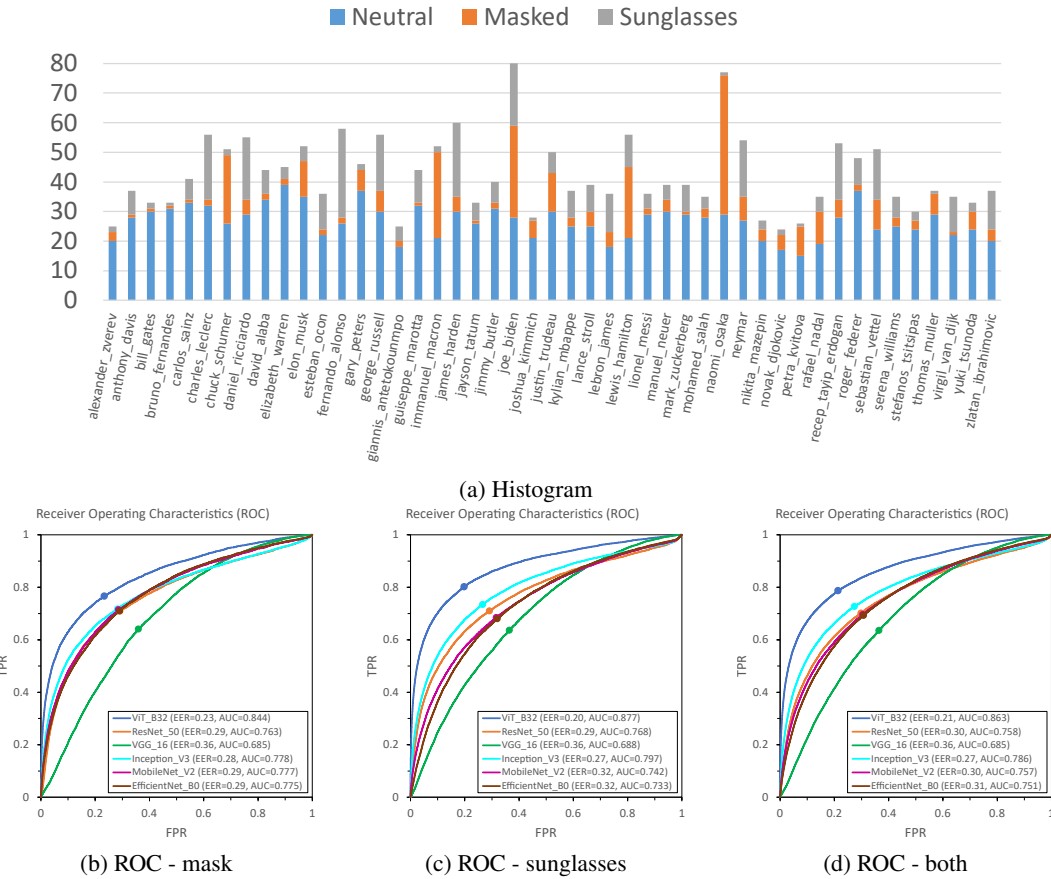

(a) Histogram

(b) ROC - mask

(c) ROC - sunglasses

(d) ROC - both

Figure 5: Histogram (a) displaying the distribution of the ROF dataset, as well as the ROC curves obtained for the categories of mask (b), sunglasses (c), and both combined (d).

discrimination and resilience against the influence of distance. This distinction becomes particularly evident at a distance of 30 meters, where CNNs' AUC scores fluctuate around 0.5, indicating random behavior, while ViT maintains an AUC score of 0.63. In the scenario involving masks, portrayed in Figure 6b, all six networks perform similarly, with ViT continuing to rank among the top performers. The complexity inherent in real-world scenarios, a characteristic captured by this dataset, contributes to the differences observed in the results at different distances, which do not decrease linearly with distance. Similarly, the mask and unmasked scenarios do not follow the exact same patterns. To gain a comprehensive understanding of both masked and unmasked scenarios across various distances, we conducted an additional experiment in which we compared all gallery images with all probe images, combining all distances. These results are illustrated in Figure 7, offering a more generalized perspective on this dataset. For the scenario without masks, results align with our earlier observations, with ViT consistently emerging as the top-performing model. Regarding the scenario with masks, ViT maintains a strong performance but ranks second to VGG, which surpasses it by a small margin, boasting a 4% increase in AUC. This occurrence is an exception; VGG has learned to detect a specific set of features that, in general, perform less effectively than ViT. However, for this particular scenario involving a small dataset, very small and occluded images due to masks, it performs better. This incident is not replicated across the other datasets used for evaluation, indicating it as an isolated case that seems to be a non-reproducible anomaly. VGG's success in this specific context can be attributed to its ability to leverage certain characteristics that may be advantageous for these particular conditions. It's noteworthy that ViT, in its overall performance across multiple datasets, consistently exhibits superior discrimination and resilience, as observed in our broader evaluation. This isolated instance highlights the complexity of real-world scenarios and the potential influence of dataset characteristics on individual model performance, emphasizing the need for robust and diverse evaluations.

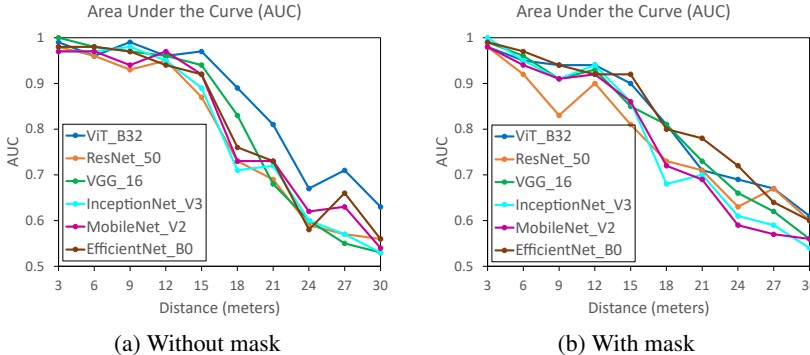

Figure 6: AUC curves derived from the ROC curves obtained at every intermediate distance in the UPM-GTI-Face dataset for the unmasked (a) and masked (b) scenarios.

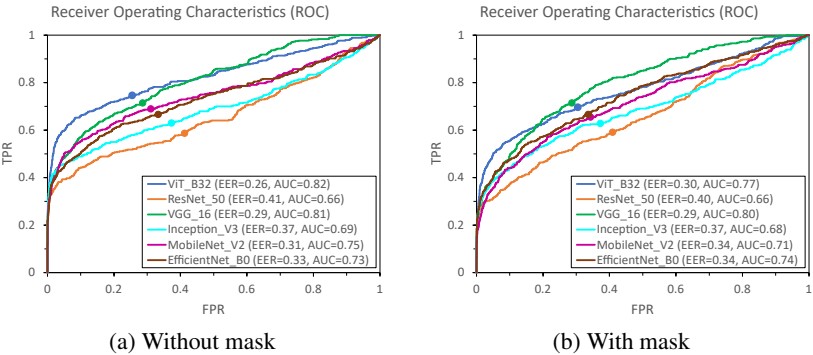

Figure 7: ROC curves obtained from combining all intermediate distances (from 3 to 30 meters) in the UPM-GTI-Face dataset for the unmasked (a) and masked (b) scenarios.

## 4 CONCLUSION

In this comprehensive study, we have conducted a thorough comparison between ViTs and CNNs in the context of face recognition tasks, including face identification and face verification subtasks. Our investigation delves into several crucial aspects.

First, we have introduced some of the primary architectural and operational disparities that set ViTs and CNNs apart. These differences lay the foundation for our extensive experimentation. Next, we meticulously detailed our experimental setup, encompassing both software and hardware configurations. This transparent framework ensures the rigor and reproducibility of our findings. Moreover, we detailed our training strategy for six distinct networks, including ViT, ResNet, VGG, Inception, MobileNet, and EfficientNet.

We subjected these networks to rigorous tests employing five diverse face image datasets, each presenting unique challenges inherent to face recognition tasks. Our meticulous analysis culminated in several noteworthy observations. ViT consistently emerged as the top performer across nearly all tests, with only a few exceptions. This underscores the exceptional capabilities of the ViT architecture in the realm of face recognition. ViT's smaller memory footprint holds the potential to facilitate the utilization of larger batch sizes, thereby promising enhanced results and swifter training times. ViT demonstrated an impressive inference speed, rivaling even the fastest CNNs, establishing its competitiveness in real-time applications. ViT's face embeddings exhibited remarkable resilience to image distortions arising from variations in camera distance and local occlusions, both in the upper and lower facial regions.

In conclusion, this paper contributes significantly by providing a comprehensive evaluation comparing ViTs against CNNs, especially in addressing challenges specific to face recognition tasks. Our findings highlight ViT's potential in this domain, shedding light on its strengths and implications for advancing face recognition systems.

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
