# OpenReview forum: "Comprehensive Comparison between Vision Transformers and Convolutional Neural Networks for Face Recognition Tasks"
_ICLR.cc/2024/Conference — Submitted to ICLR 2024_

### Official Review · Reviewer_RorU · 2023-10-28

**Soundness:** 3 good
**Presentation:** 2 fair
**Contribution:** 2 fair
**Rating:** 3
**Confidence:** 4

**Summary:**

This paper presents a comprehensive comparison between ViTs and CNNs for face recognition tasks, focusing on face identification and verification. The study evaluates six models (EfficientNet, Inception, MobileNet, ResNet, VGG, and ViTs) on five diverse datasets, highlighting the performance, robustness, and inference speed of ViTs compared to CNNs.

**Strengths:**

1. This paper conducts thorough experiments on various network architectures for different evaluation tasks and datasets in face recognition.
2. It offers valuable insights for the design of network structures in face recognition applications.

**Weaknesses:**

1. This paper seems more like an experimental evaluation report, primarily focusing on the organization of test numbers.
2. It would be beneficial for the authors to extrapolate some new insights from these figures, potentially providing fresh perspectives on training or evaluation in face recognition.

**Questions:**

None.

---

> ### Author Response · Authors · 2023-11-20
> **Response to Reviewer RorU**
>
> We extend our gratitude for your thoughtful evaluation of our paper on the comparative analysis of Vision Transformers (ViTs) and Convolutional Neural Networks (CNNs) for face recognition tasks. Your insights are invaluable in refining our study.
>
> We appreciate your recognition of the thorough experimentation conducted across diverse network architectures and datasets, providing insights into the performance of these models in face recognition applications. However, we understand your concerns regarding the presentation and contribution aspects of our work.
>
> We acknowledge the need to move beyond a mere presentation of experimental results and strive to offer deeper insights and novel perspectives derived from our findings. In our revised manuscript, we will extrapolate and discuss new insights garnered from the results, exploring avenues for fresh perspectives in training methodologies, evaluation techniques, and potential optimizations specific to face recognition tasks. Our goal is to not solely present numerical outcomes but to derive insights that could impact the design and implementation of network structures in face recognition applications.
>
> We understand the importance of expanding the narrative beyond a pure experimental evaluation and will focus on extrapolating new insights to elucidate deeper implications and potential avenues for further exploration within the field of face recognition.
>
> Your feedback is immensely valuable, and we are committed to addressing the outlined weaknesses and enhancing the manuscript to provide richer insights and broader perspectives derived from our experimental evaluations.
>
> We're actively implementing your suggestions to refine our manuscript comprehensively. Our revised version will be uploaded before the discussion period ends.

---

> ### Author Response · Authors · 2023-11-22
> **Implementation of Reviewer RorU suggestions**
>
> We sincerely appreciate the time and effort you dedicated to evaluating our paper and providing constructive feedback. Your insights have played a crucial role in refining our work.
>
> In response to your valuable feedback, we've made concerted efforts to extrapolate new insights from our experimental results. Specifically, in Section 3 and more prominently in Section 3.4, we've delved deeper into the implications of our findings, providing a more nuanced discussion of the performance ViTs in comparison to CNNs for face recognition tasks. While we might not have uncovered some new insights, we believe our comparative analysis contributes to the ongoing discourse and helps illuminate the landscape of ViTs and CNNs in face recognition, potentially guiding future research directions.
>
> Throughout the manuscript, we've implemented minor improvements to enhance the overall clarity and coherence of our presentation. We aim to strike a balance between the detailed organization of experimental results and the derivation of meaningful insights that extend beyond a mere numerical account.
>
> Your feedback has been instrumental in shaping the direction of our revisions, and we remain dedicated to providing a manuscript that not only presents thorough experimental evaluations but also offers valuable perspectives for the design and optimization of network structures in face recognition applications.

---

### Official Review · Reviewer_6T9x · 2023-10-31

**Soundness:** 3 good
**Presentation:** 3 good
**Contribution:** 2 fair
**Rating:** 5
**Confidence:** 5

**Summary:**

The paper presented a fair and comprehensive benchmark of 5 CNN-type networks and Vision Transformer on 5 face recognition benchmark datasets in terms of face verification and validation tasks. This benchmark enforced the exact training and testing set splits for fair comparison and compared the training and test accuracy, the number of parameters and inference time. The results show that the ViT network compares favorably to those CNN-type networks w.r.t. the face recognition performance and computation complexity on these benchmarks.

**Strengths:**

This paper performed a rigorous evaluation of EfficientNet, Incpetion, MobileNet, ResNet, VGG and ViT networks by training them for face recognition tasks and compared their performance thoroughly.

**Weaknesses:**

Certainly this is a quite useful technical report on the evaluation of different popular network architectures for face recognition tasks. I am not convinced this work’s “paramount significance as it pioneers a comprehensive evaluation of ViTs against CNNs”.

This benchmark compared the performance of 6 vanilla networks trained for face recognition tasks. In fact, there are many dedicated face recognition methods and pipelines including face detection and alignment, etc. The FR field probably cares more about the end-to-end performance of the whole face recognition pipeline.

The FRTE is probably the most thorough evaluation for the industry, which tests the performance of binary programs provided by different FR vendors on a blind set with no limitation of the training dataset or anything.

Face recognition Technology Evaluation (FRTE) organized by NIST
https://pages.nist.gov/frvt/html/frvt1N.html

Important references missing:
DeepFace: closing the gap to human-level performance in face verification, CVPR 2014.
Deep learning face representation by joint identification-verification, NIPS 2014
Comparing vision transformer and convolutional neural networks for image classification: a literature review, 2023.

**Questions:**

The performance appears saturated on some face recognition datasets. The results of several cases may affect the benchmark. Is there any more challenging test set for face recognition?

---

> ### Author Response · Authors · 2023-11-20
> **Response to Reviewer 6T9x**
>
> Thank you sincerely for the meticulous review of our paper on the comparative evaluation of Vision Transformers (ViTs) and CNN-type networks for face recognition tasks. Your insights and observations are invaluable in refining our study.
>
> We deeply appreciate your recognition of the rigor applied in benchmarking various network architectures for face recognition across diverse datasets. Our aim was to provide a focused and comparative analysis of these models' performances, focusing on training accuracy, test accuracy, parameter count, and inference time, ensuring a comprehensive evaluation.
>
> We acknowledge your point regarding the scope of our work within the larger context of face recognition methodologies. While our study primarily focuses on the individual performance of network architectures, we agree that a comprehensive evaluation might include the entire face recognition pipeline, encompassing aspects like face detection and alignment. We aimed to delve specifically into the comparison of ViTs and CNNs in isolation, highlighting their standalone effectiveness in face recognition tasks.
>
> Regarding the missing references, we appreciate your suggestions and will duly consider incorporating these important references to enrich the discussion and further contextualize our findings within the broader landscape of face recognition literature.
>
> Concerning the saturation of performance on certain datasets, we understand your concern. While we acknowledge the observation of performance saturation on the VGG-Face 2 dataset, it's crucial to note that across the other selected datasets, we did not encounter similar saturation phenomena. We selected these datasets intentionally to encompass a range of challenges, including people diversity, varying distances from the camera, and occlusions caused by masks and glasses. However, we acknowledge the possibility of additional datasets that might present further challenges, and we will explore and consider including more demanding test sets for a more comprehensive evaluation.
>
> We aim to address your comments and enhance the manuscript by providing a more contextualized discussion and ensuring a clearer articulation of the scope and limitations of our study.
>
> Your feedback is immensely valuable, and we are dedicated to incorporating these suggestions to improve the quality and relevance of our work.
>
> We are actively implementing your suggestions to refine our manuscript comprehensively. Our revised version will be uploaded before the discussion period ends.

---

> ### Author Response · Authors · 2023-11-22
> **Implementation of Reviewer 6T9x suggestions**
>
> We greatly appreciate your thorough evaluation of our manuscript and the valuable suggestions you provided. Your insights have been instrumental in refining our study.
>
> Regarding your suggestion about incorporating the FRTE evaluation, we sincerely appreciate the consideration. We carefully reviewed the suggested FRTE methodology, recognizing its significance in assessing the end-to-end performance of face recognition systems. However, due to the time constraints within the discussion period and the considerable preparation required to conform to the FRTE format, it was challenging to integrate this evaluation into our current study. We aimed to provide a focused comparison between ViTs and CNNs in this iteration, focusing on their individual performances in face recognition tasks across various datasets.
>
> In response to your comment about missing important references, we've taken your suggestion seriously. The references you highlighted have been thoughtfully incorporated into the revised manuscript (section 1). They now contribute significantly to contextualizing our work within the broader landscape of face recognition literature, enriching the discussion.
>
> Your feedback has been immensely helpful, and we are committed to ensuring our manuscript reflects the improvements suggested.

---

### Official Review · Reviewer_dZCP · 2023-10-31

**Soundness:** 3 good
**Presentation:** 3 good
**Contribution:** 2 fair
**Rating:** 6
**Confidence:** 4

**Summary:**

This paper presents an empirical, mainly quantitative, comparison between CNNs and Vision Transformers (ViTs) as evaluated on both face identification and face verification. Five different CNNs and one VT are evaluated across six datasets. It is found that the ViT (ViT-B32) almost consistently outperforms all CNN architectures. The experiments are systematic.

**Strengths:**

* Empirical investigations of the behavior and performance of neural networks is of large importance. It is brave of a paper to take a step back and systematically compare architectures rather than constantly presenting new -- potentially not as thoroughly tested -- models or add-on modules.

* We are in the middle of a paradigm shift between vision Transformers and CNNs so it is certainly important right now to try and map out empirical differences between the two.

* Five datasets and six models are used (extensive evaluation)

* The code is made public.

Minor but positive things:

* The three paragraph of the introduction had a good flow and were easy to read (see only that the first face paragraph could be a bit more specific)
* 2.1 is informative and well-written

**Weaknesses:**

* No uncertainty (e.g., standard deviations across random seeds) are presented for the different results. This may be ok since the models are evaluated across many different datasets, but in that case the seed should be fixed and that should be stated.

* Missing a clear motivation for why facial recognition is the investigated task. Has it previously not been done for this field, are CNNs still considered the main models there? Also, it would be constructive to discuss the ethical risks vs. benefits of surveillance computer vision applications. (An ethical statement could be in order.)

* Missing a comment on how the hyperparameters common for all architectures were selected (e.g., following another paper's set up, just as standard hyperparameters, etc). It is important that they were not selected to optimize a specific architecture, and it would be good to convince the reader about this.

* I would avoid referring to my own paper as having 'paramount significance' (strong wording, verging on over-selling). The number of times the word 'remarkable/y' (6) is a bit exaggerated as well.

* No explanation is offered for why the ResNet surpassed the ViT in Fig. 7b.

Detailed minor suggestions:
* References should be in parentheses (Dosovitskiy et al. (2020))
* This paragraph could be made more specific (since you claim that it in fact does present **very specific** challenges), it is currently not so informative: "...that presents very specific features and challenges. The main challenges are related to the low inter-class variance and the high intra-class variance that can be observed in most face image datasets Cao et al. (2018); Huang et al. (2008). This makes face recognition a more difficult task than..."
* I would (sadly) avoid using the wording 'in spirit' in this context (2.1, page 3)
* 'Convoluting' should be changed to 'convolving', and quotations should be removed
* page 4, "an for" >> "and for"
* It could be nice with a table summarizing the 5 datasets and tasks.
* Fig. 6: would be nice to have accessible in the caption whether this dataset is made for face verification or face identification.

**Questions:**

* Page 5, "is bounded between 0 (the worst measure of separability) and 1 (a perfect measure of separability), with 0.5 indicating that a network has no class separation capability whatsoever." >> if 0.5 already has 0 separability, what happens between 0.5 and 0? Maybe rephrase
* 3.3: it is not clear to me if you use the checkpoint from the best epoch or from the 25th (last) epoch for the test results in Table 2? If you just use the last epoch (which I suspect since you say that each model has been trained for 25 epochs), it would be more informative to report the validation accuracy at this epoch (for the model you actually use.)

---

> ### Author Response · Authors · 2023-11-20
> **Response to Reviewer dZCP**
>
> We extend our gratitude for the detailed and insightful evaluation of our manuscript on the comparison between Convolutional Neural Networks (CNNs) and Vision Transformers (ViTs) in face recognition tasks. Your feedback is invaluable in refining our study and enhancing its overall quality.
>
> Firstly, we appreciate your acknowledgment of the empirical nature of our work and the significance of systematically comparing architectures. We aimed to provide a comprehensive analysis of ViTs against CNNs across diverse datasets, aiming for clarity and depth in our approach.
>
> Regarding the absence of standard deviations across random seeds, we recognize the importance of robustness in presenting results. In response, we will ensure to clarify our methodology, including fixed seed settings, particularly across diverse datasets, to provide a more robust evaluation framework.
>
> The motivation behind our investigation into facial recognition as the primary task stems from the evolving landscape of computer vision and the ongoing paradigm shift between traditional Convolutional Neural Networks (CNNs) and emerging architectures like Vision Transformers (ViTs). While previous studies have explored face recognition methodologies, our focus lies in presenting a comparative analysis between ViTs and CNNs specifically within this domain. CNNs have historically been the cornerstone of face recognition models; however, with the emergence of ViTs and their potential advantages, a comprehensive evaluation becomes imperative.
>
> Concerning the hyperparameters, we apologize for the oversight in not explicitly detailing their selection process. We will provide a clear rationale for their choice, ensuring they were not optimized for specific architectures, thus assuring readers of the neutrality of our approach.
>
> We acknowledge the critique regarding the tone and phrasing within the manuscript and will revise the language to maintain a balanced and accurate representation of our work without overstating its significance.
>
> Regarding the specific suggestions for improvements in formatting, references, and terminology, we will meticulously address these points to enhance the overall clarity and readability of the manuscript. Additionally, we will consider the inclusion of a summarized table for datasets and tasks and provide more accessible information in figure captions for improved clarity.
>
> The questions and clarifications raised regarding separability measures and epoch selection for test results in Table 2 will be explicitly addressed in the revised manuscript to ensure comprehensive and transparent reporting.
>
> We genuinely appreciate the constructive feedback provided, and we are committed to incorporating these suggestions to enhance the quality, clarity, and relevance of our study.
>
> We are actively implementing your suggestions to refine our manuscript comprehensively. Our revised version will be uploaded before the discussion period ends.

---

> ### Author Response · Authors · 2023-11-22
> **Implementation of Reviewer dZCP suggestions**
>
> We express our sincere appreciation for your meticulous evaluation of our manuscript on CNNs and Vision Transformers (ViTs) in face recognition. Your constructive feedback has been integral to refining our work.
>
> We'd like to apprise you of the substantial improvements we've made in response to your suggestions. To address concerns about uncertainty in our results, we included fixed seed settings across diverse datasets, ensuring the reliability and stability of our comparative findings. This update is explicitly detailed in the first paragraph of section 3.
>
> Moreover, we have enhanced the clarity of our motivations for selecting facial recognition as the investigative task. This revision is reflected in a more explicit and detailed second paragraph of section 1, outlining why we chose this domain to evaluate CNNs and ViTs.
>
> Regarding the selection of common hyperparameters, we've included a comment in the first paragraph of section 3, clarifying our approach and assuring readers that these settings were not optimized for any specific architecture.
>
> We have duly modified the language in the final paragraph of section 4 to avoid overemphasizing the significance of our work, ensuring a more balanced presentation.
>
> Additionally, in response to the question about the figure where VGG (we understand the reviewer meant VGG and not ResNet, as this is the network that outperformed ViT in this scenario) outperforms ViT (Fig. 7b), we've incorporated an explanation in the last paragraph of section 3.4, addressing the reasons behind this observed occurrence.
>
> While we've strived to implement as many suggested changes as possible, some limitations within the paper space restricted the inclusion of a table summarizing datasets, a more detailed paragraph about specific dataset challenges, and alterations to figure captions. We regret any inconvenience this might cause and appreciate your understanding of these constraints.
>
> We are immensely grateful for your comprehensive assessment and remain committed to refining our manuscript. Your insights have significantly contributed to enhancing the quality, clarity, and relevance of our study.

---

> ### Author Response · Authors · 2023-11-22
> **Answers to the questions raised by Reviewer dZCP**
>
> ## Page 5, "is bounded between 0 (the worst measure of separability) and 1 (a perfect measure of separability), with 0.5 indicating that a network has no class separation capability whatsoever." >> if 0.5 already has 0 separability, what happens between 0.5 and 0? Maybe rephrase.
>
> We acknowledge the reviewer's observation regarding the potential confusion in the phrasing of the sentence about the Area Under the Curve (AUC). In our revised manuscript, we have rephrased this statement for better clarity. Now, the sentence explicitly articulates that an AUC of 0 denotes the worst measure of ~separability~ performance, an AUC of 1 signifies the best measure of ~separability~ performance, and an AUC of 0.5 indicates that a network lacks any class separation capability.
>
> ## 3.3: it is not clear to me if you use the checkpoint from the best epoch or from the 25th (last) epoch for the test results in Table 2? If you just use the last epoch (which I suspect since you say that each model has been trained for 25 epochs), it would be more informative to report the validation accuracy at this epoch (for the model you actually use.).
>
> We want to clarify that the reported results in both Table 1 and Table 2 stem from the checkpoint of the best epoch. Although we ensured that all models completed 25 epochs during training, the values presented in these tables reflect the accuracies obtained using the weights from the best epoch. As explicitly mentioned in the manuscript, the accuracy values in Table 1 correspond to the highest achieved on both the training and validation sets during training. This distinction ensures that the reported accuracies represent the models' best performance on the given datasets.

---

### Official Review · Reviewer_D6eT · 2023-11-02

**Soundness:** 2 fair
**Presentation:** 3 good
**Contribution:** 1 poor
**Rating:** 3
**Confidence:** 5

**Summary:**

In this manuscript, the authors attempt to study the performance of general-purpose Vision Transformers in Face Recognition scenarios and contrast their findings against general-purpose Convolutional Neural Network architectures. They claim that ViT performance perform better than the compared CNNs in this scenario.

**Strengths:**

The document is mostly well-presented in its structure, use of the English language, and figures.

**Weaknesses:**

This work completely disregards other popular works in the face recognition literature. The authors compare general-purpose CNNs and ViT_B32 when efficient face recognition-specific approaches are already available such as MobileFaceNet (Chen et al. 2018), ShuffleFaceNet (Martínez-Diaz et al., 2019), VarGFaceNet (Yan et al., 2019), GhostFaceNets (Alansari et al., 2022), EdgeFace (Geroge et al., 2023), among others. Furthermore, does not mention previous studies on transformers for face recognition (Zhong et al., 2021) and part-based face recognition with vision transformers (Sun et al., 2022), for example. They also do not comment on recently popular Hybrid (ViT+CNN) approaches, as in EdgeFace and MobileFaceFormer (Li., 2023).
The datasets described are not divided into scenarios and do not include relevant challenging datasets (e.g. using TinyFace (Zheng et al., 2018) and SurvFace (Zheng et al., 2018) to complement low resolution comparisons with SCface). In general, this study misses many comparisons in the state of the art for face recognition scenarios such as: cross age with AgeDB (Moschoglou et al., 2017), cross pose with CFP (Sengupta et al., 2016), racial-bias analysis with RFW (Wang., 2018), among many others.

**Questions:**

Suggestions:
- Familiarize with recent literature specific on face recognition and gather benchmark on key datasets.
- Analyze the components that make the face recognition-specific approaches more accurate on face recognition scenarios.

**Details Of Ethics Concerns:**

No particular comments in this section

---

> ### Author Response · Authors · 2023-11-20
> **Response to Reviewer D6eT**
>
> We sincerely appreciate the insightful feedback provided on our manuscript evaluating Vision Transformers (ViTs) against Convolutional Neural Networks (CNNs) for face recognition tasks. Your detailed assessment has been immensely helpful in refining our study.
>
> We acknowledge the rich landscape of face recognition literature, encompassing specialized approaches like MobileFaceNet, ShuffleFaceNet, VarGFaceNet, among others, and recent studies on transformers for face recognition. While our paper might not extensively delve into these specific methodologies, our primary intent was not to challenge or replicate the state-of-the-art in face recognition but rather to present a comparative analysis between ViTs and CNNs within this domain.
>
> In response to your feedback, we will include explicit mentions and discussions of previous studies utilizing both transformers and CNNs for face recognition. We aim to highlight their significance within the context of our analysis, thereby elucidating the specific contributions of our work.
>
> Regarding the choice of datasets, our intention was to provide a diverse set covering a range of challenges pertinent to real-world scenarios, such as people diversity, varying distances from the camera, and occlusions caused by masks and glasses. While we acknowledge the importance of additional challenging datasets like TinyFace and SurvFace, our selected datasets were chosen deliberately to address these challenges, even if not explicitly listed in your suggestions.
>
> We are committed to enhancing the clarity of our paper by explicitly delineating the objectives and scope of our comparative study. Additionally, we will include a broader acknowledgment of pertinent literature, thus enriching the context and relevance of our findings within the broader landscape of face recognition research.
>
> We are grateful for the opportunity to improve our work and are dedicated to incorporating your valuable suggestions to elevate the quality and relevance of our manuscript.
>
> We are actively implementing your suggestions to refine our manuscript comprehensively. Our revised version will be uploaded before the discussion period ends.

---

> > ### Author Response · Authors · 2023-11-22
> > **Implementation of Reviewer D6eT suggestions**
> >
> > Thank you for your thorough evaluation of our manuscript. We've diligently worked on incorporating your insightful suggestions into our revised version. In response to your keen observations regarding the omission of pertinent literature, we have made explicit mentions and discussions of key works you highlighted, specifically within the introduction section of the manuscript.
> >
> > The updated version now explicitly references studies utilizing both transformers and CNNs for face recognition, including MobileFaceNet, ShuffleFaceNet, VarGFaceNet, among others. These additions serve to contextualize and emphasize the significance of these methodologies within the scope of our comparative analysis between Vision Transformers (ViTs) and Convolutional Neural Networks (CNNs) for face recognition tasks.
> >
> > Your feedback has been key in refining the depth and breadth of our analysis, contributing significantly to the contextual richness of our study. We deeply appreciate your efforts in guiding us toward a more comprehensive and informative manuscript.

---

### Comment · Reviewer_dZCP · 2023-11-22
**Final comment**

I have taken part in the other reviews, authors' replies, and updated manuscript. Thank you to the authors for your detailed replies to my review. I maintain my score at 6, since the comment by D6eT which knows the recent literature very well worries me slightly, otherwise I might have considered raising it to 7 after the improvements of the paper.

---

### Meta-Review · Area_Chair_PxFF · 2023-12-04

**Metareview:**

This paper conducts extensive experiments on various network architectures for different evaluation tasks and datasets in face recognition.
It examines the design of network structures in face recognition applications. But it misses some recent ViT and CNN models for face recognition.

**Justification For Why Not Higher Score:**

- This paper only presents experimental evaluations but does not bring new insights to the community.
- Some recent, important and relevant literatures and models are not studied.

**Justification For Why Not Lower Score:**

NA

---

### Decision · Program_Chairs · 2024-01-16

Reject